# Culturally adapting a mindfulness and acceptance-based intervention to support the mental health of adolescents on antiretroviral therapy in Uganda

Khamisi Musanje[1,2]*, Carol S. Camlin[3], Moses R. Kamya[4], Wouter Vanderplasschen[5], Deborah Louise Sinclair[5], Monica Getahun[6], Hope Kirabo[7], Joan Nangendo[1], John Kiweewa[8], Ross G. White[9], Rosco Kasujja[7]

1 Clinical Epidemiology Unit, Makerere University, Kampala, Uganda, 2 Department of Educational, Social and Organizational Psychology, Makerere University, Kampala, Uganda, 3 Department of Obstetrics, Gynecology & Reproductive Sciences, University of California, San Francisco, California, United States of America, 4 Department of Medicine, Makerere University, Kampala, Uganda, 5 Department of Special Needs Education, Ghent University, Ghent, Belgium, 6 Institute for Global Health Sciences, University of California, San Francisco, California, United States of America, 7 Department of Mental Health and Community Psychology, Makerere University, Kampala, Uganda, 8 Fairfield University, Fairfield, Connecticut, United States of America, 9 School of Psychology, Queens University, Belfast, Northern Ireland, United Kingdom

* khamisi.musanje@mak.ac.ug

## Abstract

The dual burden of living with HIV and negotiating life stage changes has been identified as a contributing factor to lapsed adherence among adolescents with HIV in sub-Saharan Africa. While psychosocial support can promote medication adherence, most interventions in use with adolescents were originally developed for the general population creating a gap in appropriate support. Life-stage-appropriate, evidence-based psychosocial support interventions have been used with young people in high-income contexts, prompting interest in their use in low-income contexts. However, many interventions are less effective when implemented outside of their original settings, hence the need for modifications before implementation. We aimed to culturally adapt an evidence-based psychosocial support intervention designed to improve the mental health of young people for use among adolescents with HIV in a sub-Saharan African context and to explore the acceptability of the adapted intervention among adolescents. We engaged thirty stakeholders (n = 30) in Kampala, Uganda including psychologists, psychiatrists, social workers, HIV counselors, religious leaders and adolescent peers from December 2021 to April 2022 to modify an evidence-based intervention for adolescents. Key adaptations included simplifying the language, adding local practices, integrating locally relevant slang and stories into therapy, introducing racially-congruent visuals and cards representing emotions, and adjusting therapy materials for use in resource-constrained settings. We then tested the acceptability of the intervention in a small sample of service users using a qualitative approach. We recruited nine adolescents with HIV from a participating clinic in Kampala, delivered six 90-minute sessions of the adapted intervention across three weeks and conducted in-depth

**Data Availability Statement:** The data that support the findings of this study have been uploaded as Supporting Information files to the paper.

**Funding:** Research reported in this publication was supported by the Fogarty International Center (FIC), National Institute of Alcohol Abuse and Alcoholism (NIAAA), National Institute of Mental Health (NIMH), of the National Institutes of Health (NIH) under Award Number D43 TW011304, grant recipients: MRK and CSC. The content is solely the responsibility of the authors and does not necessarily represent the official views of the National Institutes of Health. https://www.nih.gov/ This research was supported by a grant from the National Institutes of Health, UCSF-Bay Area Center for AIDS Research, P30AI027763, grant recipients: CSC and KM. https://www.niaid.nih.gov/research/cfar-site-contacts The funders had no role in study design, data collection and analysis, decision to publish, or preparation of the manuscript. KM received salary from the National Institutes of Health, UCSF-Bay Area Center for AIDS Research, P30AI027763 grant.

**Competing interests:** The authors have declared that no competing interests exist.

interviews to assess the acceptability of the intervention. We used thematic analysis to analyze the qualitative data. The adapted intervention was perceived as acceptable among adolescents with HIV, with many stating that it helped them overcome fears, increased their self-acceptance, and gave them the confidence to make careful health-enhancing decisions.

## Introduction

HIV/AIDS has had a catastrophic effect on public health, claiming approximately 36.3 million lives globally since the early 1980s [1]. It is estimated that 38 million people were living with HIV/AIDS in 2020, with 20.6 million of these living in Sub-Saharan Africa (SSA) [1]. New HIV infections were estimated at 1.5 million people, 60% of which occurred in SSA [1], yet more than half of the new global infections occurred among young people below the age of 25 years, with SSA accounting for 88% of these infections [2]. In Uganda, approximately 1.4 million people were estimated to be living with HIV/AIDS in 2020 of which 170,000 are young people (15–25 years) [1], and accounted for 37% of all the 38,000 new HIV infections [2].

While Uganda has received global recognition for its response to HIV/AIDS [3], young people persistently lag in the UNAIDS 95-95-95 goals (95% testing, 95% treatment and 95% suppression by 2025) [4]. While many children born HIV-positive have now reached adolescence [4], new HIV infections persist through heterosexual contacts [5], creating an unprecedented burden. Thus, prioritizing care and treatment for adolescents with HIV (AWH) is a key priority [6]. Uganda's consolidated guidelines for the treatment and prevention of HIV require that all people diagnosed with HIV immediately commence with antiretroviral therapy (ART), regardless of their health status [6]. However, the success of test-and-treat strategies hinges on optimal ART adherence: i.e., maintenance of a daily dosage and receiving 95% or more of prescribed doses in a given period [7,8].

Sub-optimal adherence contributes to drug resistance, disease progression, and increased morbidity, transmission and mortality [7]. Approximately 25% of people with HIV who interrupted treatment in 2020 returned to care with advanced disease, risking death even after resuming ART [6]. Adolescents lag behind other groups in adhering to life-saving ART [9]; of the 92.5% of adolescents initiated on ART in Uganda in 2017, only 65.5% reported optimal adherence and 54.8% achieved viral suppression [10]. Furthermore, a significant proportion of HIV-related mortalities occur among adolescents [11]. For example, between 2000–2017, while HIV-related deaths dropped by 5% among children, they doubled among adolescents [4].

Beyond improved access and adherence to ART, adolescents need support to manage the dual burden of HIV and developmental changes [4]. A diagnosis of HIV has been found to result in resentment, anger, fear, and self-stigmatization among adolescents [12–14]. These factors, in turn, exacerbate difficulties in maintaining engagement in care and treatment [12]. Therefore, psychosocial care and support are critical to treatment success in this population [14].

The Uganda national guidelines on the prevention and treatment of HIV further recommend that adolescents are given psychosocial support which is developmentally and contextually appropriate for their transitional life stage [6]. However, many of the supports recommended in these guidelines were originally developed for adults [11,15], and have reduced utility and efficacy among adolescents [16]. Furthermore, interventions are not targeted towards adolescents, or tied to life stage dynamics [17], and often lack standardized delivery processes to enable replication [18], thus limiting their effectiveness [19].

A differentiated approach to care that is cognizant of developmental, treatment, behavioral and contextual factors is critical for meeting the needs of adolescents [20]. The Discoverer, Noticer, Advisor-values model (DNA-v) [21], a mindfulness and acceptance-based intervention (MABI) [22] was developed in Australia to support young people develop strengths, overcome unhelpful mental habits and self-doubt, and live fully in the present moment and make self-fulfilling choices [17,23]. DNA-v integrates attachment theory and positive psychology with mindfulness and acceptance to incorporate developmental needs in therapy [17]. Research supports the effectiveness of DNA-v in improving the mental health of adolescents although to date it has not been used with AWH [24–27]. Finally, DNA-v also fits into the behavioral skills category suggested by national guidelines [6] and can be delivered in groups by non-specialists [21], making it feasible for low-income settings where there may be few trained professionals [12,28,29].

In their original form, most psychotherapies can pose difficulties for use in non-Western settings [30]. Differences in knowledge, beliefs, values, language, and the social construction of disease, complicate the usage of Western psychotherapies, necessitating cultural modification before use in non-Western settings [31]. In contrast, culturally-adapted psychotherapies have been shown to have better outcomes outside of their original settings compared to those without adaptations [32–34]. However, a very limited number of studies have adapted evidenced-based psychosocial interventions for AWH [35]. One example is the VUKA family intervention in South Africa, which is a culturally tailored cartoon storyline and curriculum that was developed by a multi-disciplinary team of mental health providers who modified 'the collaborative HIV/AIDS and adolescent mental health program-CHAMP' based on feedback generated from stakeholder engagements and interviews [36].

The urgent need to improve treatment outcomes among AWH warrants the development of scalable and culturally-tailored interventions [6]. We sought to adapt a MABI for AWH in Uganda using a bottom-up community participatory framework [37,38]. The current paper reports on two related phases that form part of the same iterative process aimed at culturally adapting the DNA-v and exploring perceptions of AWH about the adapted intervention. Phase 1 reports on the procedure used to culturally adapt the DNA-v for use in a Ugandan context. Phase 2, on the other hand, reports on pilot testing of the adapted DNA-v to explore its acceptability among AWH using the Theoretical Framework of Acceptability (TFA) [39]. From this framework, acceptability is defined as *"a multi-faceted construct that reflects the extent to which people delivering or receiving a healthcare intervention consider it to be appropriate, based on anticipated or experienced cognitive and emotional responses to the intervention"* p.4 [40]. This paper is part of a study that aims at adapting, exploring acceptability, and evaluating the effectiveness of the adapted intervention in public healthcare centers in Kampala, Uganda.

## Materials, methods and results

The study received ethics clearance from the Makerere University School of Medicine Research and Ethics Committee (Mak-SOMREC-2021-176), the Ugandan National Council of Sciences and Technology (HS1656ES) and Kampala Capital City Authority, the administrative body that manages public health centers in Kampala.

### Setting

The research was conducted in Kampala, the capital of Uganda located in the central region, which has the highest HIV prevalence, largely attributed to urbanization and the location of the capital city [41]. In 2020, HIV prevalence in Kampala was estimated to stand at 6.2% [41].

As a central business district, Kampala serves as a destination for diverse groups of people from regions within and outside the country, making it an ideal setting for adapting and testing this intervention. Public health centers are under the administration of the city authority and offer free HIV services. Kampala is also home to the majority of the country's trained mental health providers and practitioners. Our study was embedded in two public health centers under the Kampala Capital City Authority.

### DNA-v therapy

The DNA-v is a manualized therapy with six time-limited sessions delivered to both individuals and groups to respond to the developmental needs of young people [17]. The content and focus of the DNA-v have been influenced by Acceptance and Commitment Therapy, a form of psychotherapy that combines acceptance and mindfulness strategies to create psychological flexibility [42].

All six sessions of the DNA-v center on three functional classes of behavior that promote values and consistent actions in the acronym *DNA*: *D* stands for *Discoverer* (supporting young people to explore the world around them through trial and error); *N* for *Noticer* (young people develop the capacity to allow and be aware of experiences non-judgmentally); and *A* stands for *Advisor* (building young people's awareness of the "inner" voice that is programed to problem solve). The DNA-v supports young people to become open to the existence of all three windows and, based on the situation, flexibly move from one window to another in ways that bring meaning (values) [21]. By the end of the therapy, young people develop skills to clarify values that can influence their behavioral choices, recognize that they can get caught up in their thoughts, learn to become more present with the breadth of their experience, and become able to notice thoughts and feelings as they come and go. When using the DNA-v, contextual awareness is key as it influences the perspectives young people form about themselves [17].

We selected the DNA-v for adolescents because it has been demonstrated to improve young people's mental health [17,22,24]. In a controlled trial conducted among adolescents in the USA, the DNA-v group reported better sleep hygiene and physical activity [24], Improved mental health is a key facilitator of adherence among people with HIV [14,43]. Furthermore, interventions grounded in mindfulness and acceptance incorporate elements of mindfulness to help people be more present in a non- judgmental way to their current experiences which can promote emotion regulation, and counteract the imbalance of developmental changes amidst under-developed executive functions such as judgment [44]. The World Health Organization has also used mindfulness and acceptance principles to develop Self Help Plus (SH+) a group-based psychosocial intervention that has been adapted for use in a variety of humanitarian settings including in Uganda [45]. Additionally, the authors of the current paper are familiar with the principles upon which DNA-v is built and also had knowledge about the local context and judged it to be appropriate for use in the context. Finally, its ease of use, flexibility in delivery, usage in groups, availability of an existing manual, and potential for scalability make the DNA-v approach a potentially important one for use in a low-resource setting.

We present the methods and results of phase 1 (adaptation) and phase 2 (testing) separately, before jointly discussing the implications of the two phases.

### Methods

### Phase 1: Culturally adapting the DNA-v for use in Uganda

**Cultural adaptation approach.** Several models and frameworks have been proposed to guide the adaptation of psychotherapies [30,46,47]. We drew on the Ecological Validity Model (EVM) [46], due to its wide use in adaptation initiatives [48–50]. The EVM proposes eight

components for adaptation [46]; we utilized six components adopted in an earlier study with refugees in Uganda [51], namely: (i) the social context, (ii) methods of delivery, (iii) negotiating differences in knowledge, values and practices, (iv) goals, (v) group relationships and (vi) language and metaphors [51]. Since EVM provides no clear guide on steps for adapting interventions, we limited its usage to the identification of adaptation components and then combined it with the Formative Method for Adapting Psychotherapy (FMAP) [37] as a guide to the adaptation process. FMAP utilizes a bottom-up approach with stakeholders and suggests five steps for adapting and testing psychotherapy. These steps include: (i) generating knowledge and collaborating with stakeholders; (ii) integrating generated information with theory and empirical clinical knowledge; (iii) reviewing the initial culturally adapted clinical intervention with stakeholders and revising the adapted intervention; (iv) testing the culturally adapted intervention; and (v) synthesizing knowledge produced by earlier phases and finalizing the cultural adaptation [37].

**Participants.** We recruited n = 30 participants across eight categories of stakeholders; (i) counsellors working with adolescents in ART clinics (n = 5), (ii) child and adolescent psychologists (n = 4), (iii) educational psychologists'/ school counselors (n = 3), (iv) psychiatrists (n = 2), (v) spiritual leaders (n = 3), (vi) social workers (n = 3), (vii) graduate clinical or counselling psychology students (n = 3) and (vii) AWH (n = 7). The median age was 37 years, (SD = 10.4), with over half being females (n = 18). All participants were proficient in English and Luganda.

We leveraged the network of local organizations to recruit participants, including the Uganda Counselling Association, Uganda Clinical Psychology Association, Makerere University, psychiatrists through Butabika Hospital (a national referral psychiatric hospital), as well as HIV care providers and social workers recruited through the Infectious Disease Institute—a local non-profit research organization. Local mental health experts were purposively recruited based on their experience of working with AWH and/or knowledge of group psychotherapy, while convenience sampling targeting balance on gender and age was used to recruit AWH through the Network of Young People living with HIV/AIDS. Spiritual leaders were identified by adolescent peers. All participants received information on the study and provided written informed consent and assent (for minors) before recruitment into the study. Although the consolidated guidelines for the prevention and treatment of HIV in Uganda encourage AWH above 12 years of age to seek care without parental consent [6], we sought parental consent as a procedure recommended by the organization from which we recruited participants.

**Procedure.** Our adaptation and testing procedure is outlined in Fig 1.

1. *Generating knowledge and collaboration with stakeholders*
   KM and RS who are both psychology graduates and trained in using the DNA-v facilitated two stakeholder workshops to begin the DNA-v adaptation process. The first workshop, held with local mental health providers and experts (all stakeholders except AWH), included: a (i) review of the DNA-v manual (an English version of the manual that was emailed to participants as preparatory reading a week in advance of the workshop); (ii) a group-facilitated review of the manual and other developed material during the first day of the workshop; and (iii) small group-facilitated discussions to assess the appropriateness of DNA-v, guided by the six domains of the EVM. Discussions lasted for four hours (with breaks). All discussions were conducted in English, recorded and later transcribed. We repeated a similar process with AWH one month later in a separate workshop to ensure their confidentiality and comfort with peers. The discussion with AWH was conducted in both English and Luganda and participants had the option to interact translingually. The discussions lasted for three hours (with breaks to allow adolescents to refresh). In total, we

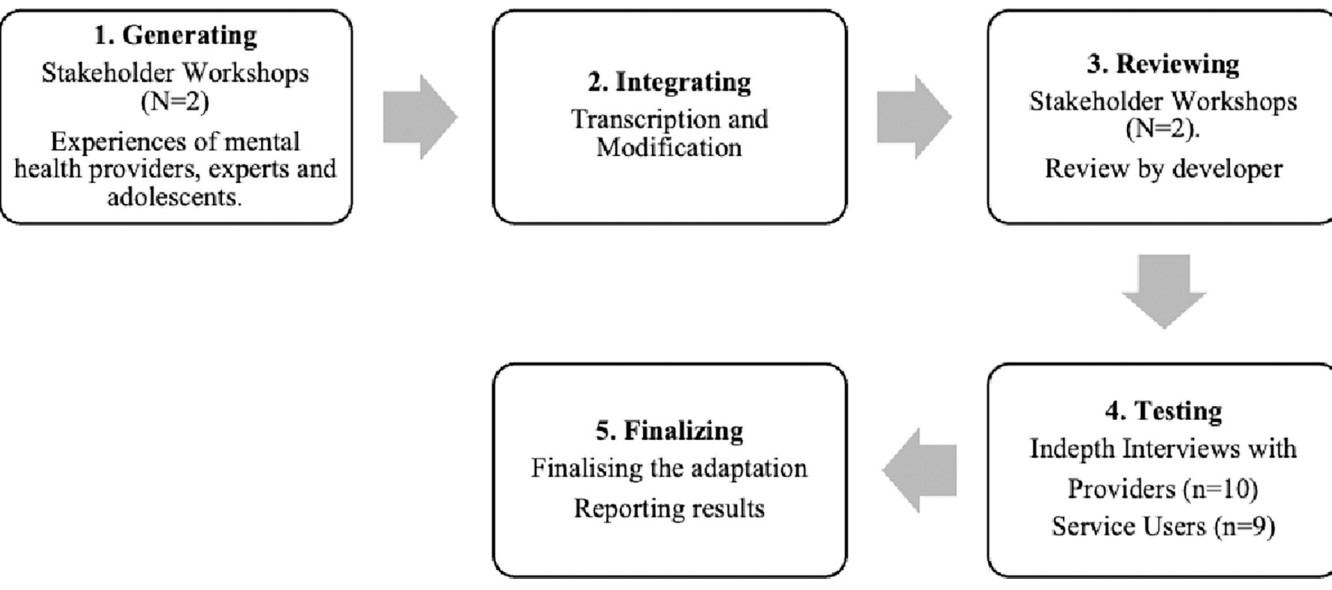

**Fig 1. Intervention adaptation process.**

conducted two workshops at the generation stage, one with local experts and another with AWH.

2. *Integrating generated information into theory and empirical clinical knowledge*
Following the workshops, we synthesized the information gathered from stakeholders with empirically supported literature on the adaptation of psychotherapy, and HIV care and treatment literature. We present details of the adaptations made later in this paper. Small group discussions during stakeholder workshops helped to maximize the involvement of attendees, minimizing the dominance of a small number of voices. We used group consensus when identifying modifications for integration. Suggestions across groups were compared to identify inter-group consensus and most recurring ideas were given top priority. We then calibrated the ideas against the norm of practices identified by the literature, before integration. In cases of inter-group contradictions, ideas were further discussed during the review stage until consensus was achieved. Two senior researchers, RW and RK, supervised the integration process. These researchers are trained in mindfulness and acceptance therapies and are experienced in intervention adaptation in Uganda.

3. *Reviewing and revising the initial culturally adapted intervention with stakeholders*
We conducted an initial review of the adapted manual following the six constructs of the EVM with five local experts that had direct experience of working with AWH, who were purposively selected from the bigger stakeholder group. A small group allowed for an in-depth review of adapted material. We provided reviewers with the adapted manual a week before the expert panel meeting to acquaint themselves with the changes. During the review, KM and RK presented the modifications and the rationale underlying the changes. We conducted a similar process with all the AWHs who participated in step 1. We then conducted a second round of refinements to the intervention manual and materials. Further, we shared the adapted manual with the developers of the DNA-v (Joseph Ciarrochi and Louise Hayes) for review and feedback.

**Analysis.**   Before the stakeholder discussions, pre-determined criteria were identified to guide how consensus would be reached in the decision-making process. A majority vote

approach was used to decide upon the key adaptations to be made [52]. We held discussions in small groups of five people who were tasked to come up with a group position on every modification item that arose from their deliberations. The research team used the most recurrent themes across groups, to synthesize with literature to inform adaptation.

## Phase 2: Pilot testing of the culturally adapted DNA-v

This phase of activity explored the acceptability of the adapted DNA-v among selected service users (AWH 15–19 years) at one of the participating health centers located in the Rubaga division in Kampala (Kitebi Health Centre III). The health center has approximately 170 AWH aged 15–19 years. Sessions were conducted between April and May 2022. We followed the guidance from the TFA to explore the acceptability of the adapted DNA-v. TFA proposes seven constructs to consider when testing the acceptability of health interventions [40]: (i) Perceived effectiveness: the extent to which the intervention is perceived as likely to achieve its purpose; (ii) Affective attitude: how individuals feel about the interventions; (iii) Self-efficacy: participant's confidence that they can perform the behavior(s) required to participate in the intervention; (iv) Intervention coherence: the extent to which participants understand the intervention and how it works; (v) Ethicality: the extent to which the intervention has a good fit with an individual's value system; (vi) Burden: the perceived amount of effort required to participate in the intervention; and (vii) Opportunity cost: the extent to which benefits, profits or values must be given up to engage in the intervention [39].

We adopted the seven constructs of the TFA to inform items in the interview schedule in a way for AWH to respond while reflecting on the appropriateness of the content, process, instructions, examples used in the intervention, and how they felt impacted by the intervention. Sample questions included: (i) How do you feel after going through these training sessions? Why do you feel this way? Is there anything that caught your attention most or confused you? What exactly caught your attention or confused you? (ii) How easy or hard was it for you to understand things we have looked at? Why? The interview schedule was piloted with a group of AWH identified from the Uganda network of young people with HIV/AIDS before data collection.

**Participants.**   We worked with healthcare providers to recruit AWH (n = 9) from a group of older adolescents attending care at the study clinic (over eight interviews are considered sufficient for saturation) [53]. A convenience sample of participants was recruited from the clinic but maximumly varied by gender, age, and period spent on ART. All participants approached agreed to participate. The median age of participants was 17 years (SD = 1.53), participants were split almost evenly by gender (females n = 5 and males n = 4) and the majority had spent more than five years on ART (n = 7). All participants were proficient in both English and Luganda. We obtained written consent from participants above 18 years of age and both parental consent and participant assent for minors. Consent forms were in both English and Luganda.

**Procedure.**   During recruitment, participants were oriented about the study, particularly the study goals, duration, their rights and expectations and reasons for conducting the research. We also used this stage to establish rapport with study participants. After recruitment, we delivered the six 90-minute sessions of the adapted DNA-v to a group of AWH (n = 9) across three weeks at the clinic premises. Since all had completed the training, we conducted in-depth interviews (IDIs) with nine AWHs. Interviewers introduced themselves and explained the reasons for conducting the study before starting. IDIs lasted between 25–30 minutes and explored the acceptability of the modified DNA-v intervention from the AWH perspective. The first author, who has extensive training in using the DNA-v, and the third

author, who is trained in mindfulness and acceptance therapies, conducted the training. At completion, three members of the research team (two males and one female), psychology graduates, trained in mindfulness and acceptance therapy and experienced in collecting qualitative data, who did not facilitate the training, conducted individual in-depth interviews with the training recipients using a semi-structured interview schedule. Using other members of the research team rather than the trainers to conduct the interviews was done to reduce socially desirable evaluations of the intervention [54]. Audio recordings from all IDIs which were predominantly in Luganda were transcribed and translated into English by two graduate students proficient in English and Luganda. Participants were assigned pseudonyms during the analysis and reporting of results.

**Analysis.** We used a thematic analysis approach [55,56] to organize findings according to *a priori* thematic categories that aligned with the themes of the theoretical framework of acceptability-TFA [39], with the use of NVivo software (version 12).

Specifically, data were analyzed using abductive analysis; a hybrid of inductive and deductive thematic analysis [57]. Two authors from different disciplines (EK) and (KM) conducted the data analysis. Both authors listened to two of the audio recordings to become familiar with the data before analysis. The authors independently used open coding on three transcripts to develop codes that represented themes that were present across responses before they met to discuss, compare labels, and agree on a core set of codes to apply to all subsequent transcripts. They, however, remained open to including other codes that would emerge as the coding process proceeded. They further reviewed and discussed sample quotes to ensure that code definitions were consistent and appropriately applied to participants' responses. In the next step, the codes that had been inductively generated were iteratively grouped into themes that were deductively developed based on the constructs of the TFA (perceived effectiveness, intervention coherence, self-efficacy, affective attitude, burden, ethicality and opportunity cost) to form an analytical framework. The subsequent transcripts were then indexed using the analytical framework, once themes were finalized writing up of the results began. To provide an illustrative example of our analysis process, codes such as 'takes time', 'discomfort with language' and 'financial costs' emerged from the inductive analysis. Using a deductive approach, these codes were grouped under 'Burden' as an a priori theme adopted from the TFA.

## Results

### Phase 1: Adaptation

We made several adaptations to the DNA-v intervention. Changes primarily involved simplifying the manual into plain English (British level A2 or basic level), replacing technical language with colloquialisms, adjusting the intervention to fit into a resource-constrained health care system, and revising the descriptions of *values*. They also involved finding locally relevant examples and adding cards depicting emotions which are used to start conversations during sessions with AWH. The goals of the DNA-v were also aligned with the needs of AWH. Key aspects of the adaptations are presented in Table 1.

**Language.** Informants observed that the manual was primarily written for native English speakers. The initial suggestion was to translate the manual to Luganda, the most widely spoken language in the central region. However, the presence of over 50 languages in Uganda made translation unfeasible. Informants agreed to use basic/plain English (British Level A2 English) locally considered as 'primary school level English'. A native speaker of English familiar with DNA-v reviewed the language changes. Examples of changes are shown in Table 1. Additional adaptations to increase the acceptability of language included using indirect, rather than direct, speech, as well as the use of requests rather than directives. For example, *'any*

**Table 1. Key adaptations to group DNA-v therapy using the ecological validity framework.**

| EVM Domain | Operationalization | Before adaptation Example | Issues raised | Adaptation decision | Suggested changes Example |
|---|---|---|---|---|---|
| Language | Translation into the local language | The manual targeted native English speakers. Words/ phrases such as: (i) Blurt stuff out (ii) Impulsive | English words used are at times difficult. Uganda also has many dialects making it hard to satisfy such a linguistically diverse population. | Simplifying the manual to plain English. | Manual in plain English (i)Blurt stuff out = Speak without thinking (*obutasooka kulowooza nga oyogeera*) (ii)Impulsive = Rushing to decide (*okupakuka okusalawo*) |
| | Use of local idioms | (i) Advisor (ii) Homework | (i)Misinterpreted as seeking advice elsewhere. (ii) As homework was associated with school, it carried a cognitive burden | (i)Replace with commonly used terms connecting to the same meaning. (ii)Use a less burdening term | (i)Advisor = The inner person or voice in the head (*omuntu alimunda*) (ii)Homework = Home practice (*byosoboola okukola nga oli waka*) |
| | Technical terms | (i)Present moment awareness, (ii)Unhook from the advisor | Difficult to understand by non-specialists. | Change to terms that lay counselors and adolescent peers can understand. | (i)Here and now (ii) Without the influence of the inner voice. |
| | Communication style | Direct style: 'what makes a good group?' | Creates tension and unease. | An indirect and non-commanding style for a culture where open and direct communication is not often used (high context culture) | The language style changed in the manual to now read: 'Suggest the qualities of a good team'. |
| Metaphors | Materials with cultural and contextual relevance | YouTube videos e.g. 'fitting in', https://www.youtube.com/watch?v=LwNJZUZFt-U 'baby responding to evil laugh' https://www.youtube.com/watch?v=YemitZJBT1Y | Technological devices and internet connections are not as freely available, affordable and thus non-sustainable in a low-income context. | Use local examples, skits, or stories that bring out the intended meaning without any financial expenses | (i)Dropping out of school when advised by friends. (ii)Using polyethylene sacks to act out 'fitting in'. Participants wore tight polyethylene sacks formed into dresses and asked to walk faster. Aim is to show the struggle associated with trying to fit in. |
| | | Values cards with predominantly white faces | Pictures with which participants could not identify, as they mostly depicted the "caucasian population" or other races. | Photos were replaced with those depicting black faces and bodies and familiar words. | Value cards depicting people that look more like those using the materials in Uganda, and simplified narratives. See supplementary materials. |
| | Proverbs, stories and local examples. | (i) Seaweed illustration (ii) Free hugs in Santori | (i)As there is no sea in Uganda, this was difficult to relate to. (ii) Hugs are culturally and religiously divisive. | (i)Use familiar illustrations (ii) Use culturally acceptable illustrations | (i)Papyrus in the swamp (ii)Tap on the shoulder, handshake or kneel as a sign of respect. |
| Differences in knowledge, values and practice | Incorporating local practice into therapy | Sessions begin with recaps and aims. | It is unusual to start a group activity without prayer. | Include prayer at the start and end of sessions. It's a common practice in Uganda. | 'Can someone lead us in prayer before we start today's session?' or 'how should we start? ' |
| | Addition of phases/ principles | Under 'Noticer', Participants share how they feel. | Many Ugandans are reluctant to be expressive in formal settings. | Spend time talking about the expression of emotions and opening up about exactly what one feels. For example, sad, excited, or worried. | Emotional cards were added. 'Sorting through these emotional cards, tell us how you feel today'. |
| | | Universal values are listed on cards: 'Being Independent' 'Connecting' 'Kindness' | AWH might have values and goals that are specific to their condition. | Add values suggested by AWH. | Values suggested by AWH: Taking care of my health, accepting myself, maintaining health-promoting routines, and not being judged. |
| | | Logical teaching of mindfulness: 'Aware-Notice-Describe' | No previous experience; mindfulness is a new process to AWH | Use day-to-day experiences to cultivate mindfulness. They should be simple and common. | Experience of eating a banana; the scent and taste. Or becoming more fully present with the journey than wanting to be at the destination when moving in a taxi? |

(*Continued*)

**Table 1.** (Continued)

| EVM Domain | Operationalization | Before adaptation Example | Issues raised | Adaptation decision | Suggested changes Example |
|---|---|---|---|---|---|
| Methods of delivery | Location of group delivery | At school or therapist clinic/office | Not all AWH are in school, and there are privacy concerns and limited access to therapists. | Prioritize confidentiality of status, use contexts recognized by the national guidelines and not too costly. | Delivery at public health care centers during clinic days. |
| | Structural adaptation | Six sessions of two hours each. | School and work schedules may not allow participation. | A flexible and sustainable schedule be used with shorter sessions. | Six sessions of 90 minutes each, with additional home practice. |
| | Adaptation of training | Facilitator-led, classroom setting and cognitively tasking | Living with HIV is already burdening. Support should be less onerous. | Consult young people on what they prefer before implementing. | Some suggestions by adolescents: acting out sessions in drama clubs and singing in funny voices to mimic the advisor. |
| Group processes | Group dynamics | Mixed groups with no clear group size. | Age, group size and trust are key aspects of group therapy for ALWHA, | Separate groups by developmental stage, keep size between 6–12 and notify participants about membership of the group before starting so that anyone uncomfortable opts out early. | Young people 14 years and below are in a separate group from those aged 15–19 years. Pre-existing groups at the clinic are considered to maintain group cohesion and age categorization |
| | | Young people come up with their own confidentiality rules. | HIV is highly stigmatized in Uganda; confidentiality must be emphasized. | Facilitators cannot leave rules of group confidentiality to ALWHA. | A confidentiality statement was added to the manual. "Whatever is shared in the group **must** remain confidential. Do you all agree?" |
| Goals | Clarifying goals | The therapy aims to help young people develop psychological flexibility. | "Therapy" is associated with clinical conditions and the concept of illness. | Reframe to 'life skills training' and clarify improving wellbeing and strengthening relationships as aims. | The new goal was added to the manual. 'The goal of this life skills training is to support you to live well, form better relationships and become who you want to be'. |
| Social context | Feasibility | Use videos, audio, and other materials in a facilitated room. | Public health centers are constrained by space and other resources. | Adjust materials for use in open spaces. Include materials that require less or no technological resources. | (i) Manila papers and posters were used for sessions delivered in a tent. (ii) Revised categories of facilitators to include adolescent peer leaders. |
| | | Facilitators are trained in DNA-v, group facilitation and adolescent therapies. | Knowledge of HIV is a requirement and well-qualified facilitators are limited. | Train onsite counselors and peer leaders at clinics to be facilitators as the first priority. | Facilitators should know about HIV, be trained in DNA-v and be available at the clinic. |
| | | Therapy is organized as an independent intervention. | Resources to facilitate independent sessions outside clinic programs are not available. | Integrate DNA-v work in the HIV healthcare system for sustainability. Counselors should be able to use it as an additional technique in daily practice. | Conceptualization forms and cards are used as tools for establishing rapport in daily counselling practice. |

suggestions on how we can make a good group?' replacing 'what makes a good group?' was considered more appropriate and respectful in the Ugandan context.

**Metaphors.** Local mental health experts found most symbolic representations in the DNA-v manual culturally and contextually incongruent. Illustrations such as 'Notice the Noticer game' were thought to be hard to understand. Further 'random free hugs in Santori' (a metaphor for creating sensations and feelings) were felt to be culturally unacceptable, with computer and internet-aided videos also impractical in resource-constrained settings. Further, the values and strength cards with White individuals in the images were perceived to be unrelatable. Informants suggested the use of local stories, trending events, songs, and familiar images on cards. We introduced a story of a person who grew up in the slums, and later became a popular presidential candidate; further, we supplemented the story with a game we

named *'life sacks'*. This game involves wearing a polyethylene sack and trying uncomfortable and rapid movements. This exercise was intended to show adolescents that they do not have to struggle to fit in. Its chief message was that it is possible to change one's life regardless of the situation, which is a central message in DNA-v [21].

We included a popular nursery school song "we began as goats" (*'twajja tuli mbuzi'*) to depict how words do not entirely define who we are, which is an important step in cultivating mindfulness and acceptance [23].

**Negotiating differences in knowledge, values and practice.** Local mental health experts compared and contrasted several of the assumptions of DNA-v to the socio-cultural realities in Uganda. Key comparisons included emotional expression, values, and cultivating mindfulness. DNA-v assumes an open expression of emotions with direct requests such as: '*tell me how you feel'* as a way of accessing deep-seated feelings. Local experts observed that Ugandans are reluctant to be expressive in formal settings, have not been nurtured/socialized to communicate emotions openly, and tend to regard expressing negative emotional reactions as a weakness. We included sections in the DNA-v manual relating to session two (Noticer) to specifically address concerns about the expression of emotions and added visual cards depicting photos and words of various emotions.

Local mental health experts who were part of the review panel at Phase 3 also felt the values statements in the DNA-v manual were generic; living with HIV is a unique and life-changing experience, and thus, generic values might not fully represent what is meaningful to AWH. Experts recommended engaging AWH to identify additional health-related values before implementation. AWH identified *taking care of my health*, *accepting myself*, *maintaining health-promoting routines* and *not being judged* as important values. The research team used this information to create additional value cards and also added a stepwise narrative on cultivating mindfulness built around simple and familiar practices such as eating a banana and blowing up a balloon, to address critiques of the direct approach included in DNA-v.

**Methods of delivery.** The EVM underscores the importance of location when adapting group psychotherapy. The DNA-v is designed as a school-based program, addressing participants as students and utilizing school-based scenarios as success case vignettes. Both experts and AWH did not find schools to be an appropriate environment for interventions targeting AWH. First, AWH revealed the stigma they experience at school from teachers and students. Second, local experts noted that a large number of AWH drop out of school, limiting the potential impact of a school-based intervention.

We noted the recommendations to deliver the intervention through health centers and adjust therapy to fit the conditions of AWH. Local experts suggested reducing the session time from 2 hours to 90 minutes and utilizing existing settings at clinics if uptake of the intervention is to be facilitated. We made the following adaptations: first, we changed the references from "students" to "adolescents". We then adjusted the session time to 90 minutes but added extra plays suggested by adolescents such as "cards laid face down on the ground" (*Biggula*) to represent the challenge of choosing (a DNA-v process), replacing lengthy narratives.

**Group processes.** We expanded the criteria for group composition beyond what was identified in the DNA-v manual. Age, group size, confidentiality, and trust were key areas for consideration when forming groups. Local experts suggested grouping adolescents by development stage (e.g. 14 and below and 15–19 years) and keeping the group size between 6–12—the typical size for group therapy in this setting. The DNA-v manual does not address group composition; thus, we added a section in the introduction to describe suggested group compositions and emphasize confidentiality to limit stigma and marginalization. We further modified the group-generated confidentiality tenets described in DNA-v, by assigning the facilitator (group therapy lead) the responsibility to emphasize confidentiality and facilitate group

agreements. We also included instructions to contact the facilitator in the event of any discomfort or breach of confidentiality.

**Goals.** Adolescents indicated that having good relationships with family and friends, and getting accepted are things they would love to see in their lives. We expanded the goal of DNA-v to incorporate these suggestions [Table 1]. We also revised the labeling of the intervention from *therapy* to *life skills training program* because AWH associated the word 'therapy' with medicines, as it is often used by providers when one is failing on treatment.

**Social context.** We included adolescent peer leaders as guided group facilitators. Informants raised concern about the DNA-v creating extra work for overburdened providers and suggested the inclusion of peer leaders. Thus, we revised the definition of a facilitator in the manual to be inclusive of peer leaders: "*a facilitator should either be; a counselor, or adolescent peer leader (guided by a counselor), having knowledge about HIV, trained in using DNA-v manual, and having prior exposure to group facilitation*". We also revised the list of training materials to remove technologically powered aids such as computers and replaced them with traditional training aids such as manila paper, markers and stickers. This adjustment aimed at addressing space and technological constraints.

## Phase 2: Testing

This phase of the study has been reported based on the consolidated criteria for reporting qualitative research (COREQ). The findings are presented in the order of the themes following the TFA constructs. The description of themes is preceded by the characteristics of the participants.

**Perceived effectiveness.** When reflecting on the extent to which the DNA-v intervention was likely to achieve its purpose, AWH expressed optimism that the DNA-v can support adolescents who interrupt treatment to improve adherence by refocusing them on what is most important in their lives. Sherry [17- year-old female AWH] captured this sentiment thus, "*In case like when a girl gets a boyfriend, they don't want them to know their status so they leave medicine. But when you bring them in this study, of course, they'll know that their life is most important.*" AWH also reported that the DNA-v helped them to develop the confidence to face the fears surrounding their life. Anna [19 –year-old female AWH], for instance, noted that "*I should start facing some of my fears, it helped me to start seeing things again in a different perspective*". By developing the confidence to face their fears AWH felt empowered to make careful decisions about their lives as stated by Tash [17-year-old female AWH] who said, "*It helps you make decisions. It helps you forego some things the head always tells you, then you say, I am sticking on this*".

Besides, AWH expressed that DNA-v engendered self-belief and gave them the confidence to deal with the negative judgment they always experience. Smart [16-year-old male AWH] typified this sentiment when he noted, "*At the garage where I work they despise me a lot because I am very small which makes me feel bad. This training has taught me to believe in myself.*" Some of the AWH linked gaining self-belief to the ability to make better life decisions as noted by Tash [17-year-old female AWH]: "*I now know how to use my body, I know how to let go and I know how to focus on what I want.*" Furthermore, AWH indicated that besides learning how to deal with painful experiences in their lives, the DNA-v allowed them to open up to inner feelings. Portable [16-year-old female AWH], noted thus: "*By the time I came here, I had anger because of how I am treated at my uncle's home. A very big thing has been taken out of my situation.*" Similarly, Jay [18-year-old male AWH] noted that "*Even though you are at the point of hating yourself, this brings you to speak how you feel at heart.*"

**Affective attitude.** Relating to how participants felt about the DNA-v, AWH found the DNA-v to be important in addressing most of their life challenges as noted by Anna [19 –year-

old female AWH] who said, "*I found it important because when you look at it, you find yourself tackling on each and every aspect of life.*" In addition, some participants noted that going through the DNA-v normalized experiences in their lives. This was the case for Portable [16-year- old female AWH], who noted, "*I liked the training because there are things I thought I was only one facing them, I've discovered that even other people go through it.*

**Self-efficacy.** Under this construct of the TFA, we report on AWH's willingness to use the skills gained from the DNA-v. AWH expressed confidence in using skills from the DNA-v as alluded to by Portable [16-year-old female AWH] who said, "*I am someone who has always been making abrupt decisions, but now I hold on a bit.*" Furthermore, AWHs who double as peer leaders expressed willingness to use the DNA-v to support fellow adolescents including those who are not living with HIV as noted by Jay [18-year-old male AWH] when he said, "*I'll use it for my fellow adolescents even though they not on ARVs, these are the ones we hang out with most, they need it too.*"

**Intervention coherence.** We looked at the ease with which AWH understood the content of the adapted DNA-v model. AWH found the DNA-v easy to understand since it relates to their lived experiences, as stated by Jay [18-year-old male AWH], "*It was easy for me to understand because all that they were teaching I had ever experienced,*" while the use of day-to-day examples made practice easy for Tash [17-year old female AWH] who noted, "*It is in a simplified language and uses the normal things, they are easy to do*".

**Ethicality.** In assessing the congruency between the goals of the DNA-v and AWH values system, participants reported that the intervention promoted self-acceptance which is an important value in their lives as AWH. Smart [16-year-old male AWH], in this regard, observed "*It causes you to accept yourself, because if you don't, you can't overcome anything.,*"

**Burden.** Relating to effort and time required to participate, AWH reported having invested a lot of time to complete DNA-v sessions, but were optimistic that the investment was worth it, as stated by Jay [18-year-old male AWH] when he said, "*It has taken some time but it was not a concern to me because I saw that this thing concerns me as a person from when it begun*". Furthermore, some expressed discomfort with parts presented in English even though sessions were a mix of both Luganda and English, as stated by Smart: "*At first it wasn't easy for me to follow properly when English was used but I kept trying*". Others shared concerns about the distance they had to cover to come to the clinic, the frequency of visits, and how they would be able to meet transport costs in case there was no transport refund. Anna [19 –year-old female AWH] noted, "*First of all, I don't live very near, I live in Makindye and it's a bit far, I spend a lot on transport and it's like we have to attend many times.*"

**Opportunity cost.** Finally, we looked at possible benefits AWH will have to forego to be able to participate in the DNA-v sessions. AWH indicated that they had to miss school, work, and other activities to complete sessions and suggested a weekend program to be more appropriate. Many shared Anna's [19 –year-old female AWH] sentiment when she said, "*I have found it a bit challenging because I had to attend this and again school at the same time. Saturday is a better day because this is also important.*" Pat, [19-year-old female AWH] agreed thus, "*I stopped all what I had to do then I decided to come to see what you're telling us*".

To complete the adaptation process, we synthesized knowledge produced from the two study stages and made a final round of refinements to the intervention where necessary as presented in Table 2.

## Discussion

Our study describes the use of a systematic and comprehensive approach to adapt an intervention to support AWH. We worked with key community members in the HIV care continuum

**Table 2. Additional refinements to the DNA-v after the testing phase.**

| EVM Domain | Adapted item in Phase1 | Feedback from Phase 2 | Suggested changes/refinements |
|---|---|---|---|
| Language | Simplifying the manual to plain English | Even with the use of plain English, some AWH had difficulties with the language. | Facilitators are encouraged to make direct translations of the DNA-v processes. Local words and slang suggested by AWH were added to the manual such as "wise up" (*kwebereramu)* bringing attention to the choices available when stuck, and "game over" (*kaweddemu)* representing a state of hopelessness adolescents encounter once their HIV status is revealed |
| | Advisor as a DNA-v process to be referred to as the "inner person" | Adolescents showed a greater understanding of the message when the reference was made to the "inner person" | We proposed to change the name in the adapted manual from DNA-v to DNI-v where "I" represents the inner person. |
| Delivery | Sessions are to be delivered on clinic days to reduce the costs of transport. | Clinic days are full of activities, happen on weekdays, coincide with work and school schedules, and limit the time available for sessions. | We recommended the adoption of a flexible schedule which does not compromise school and work but is also mindful of logistical challenges. We encouraged flexibility in using the DNI-v by introducing its principles in daily practice. |

in a cyclical iterative process to combine knowledge about intervention materials with social and cultural relevance. Key adaptations included simplifying the language, adding local practices, integrating locally relevant slang and stories into therapy, incorporating values cherished by AWH, introducing racially-congruent visuals and cards representing emotions, and adjusting therapy materials for use in resource-constrained settings. We discussed and justified contextual additions to the DNA-v while complimenting its processes and goals. Our study is the first to provide a stepped bottom-up process for adapting psychotherapy to fill the gap of developmentally and culturally appropriate support interventions for AWH in Uganda.

Contextually adapting psychotherapy helps to extend evidence-based support to contexts where the development of home-based interventions may not be feasible due to several constraints, and also improves both the implementation and clinical outcomes of such interventions when evaluated [36,58]. Our study demonstrates how contextually informed modifications to the DNA-v were well received by users in the qualitative interviews we conducted. This is consistent with studies that report benefits such as improved uptake and effectiveness that result from adapting psychotherapy for use with young people in low-income contexts [36,59].

Involving mental health service providers working directly with AWH in adapting psychotherapy contributed first-hand stakeholder experience which is important to the wide-scale adoption and uptake of interventions. This is consistent with Corburn (2003) who noted that local knowledge generated by stakeholders results in low-cost implementation strategies and eases the identification of possible implementation barriers [38]. Several studies also suggest that implementing interventions in their original form without consideration of cultural and contextual factors can limit their use and effectiveness [30,34]. Further, detaching psychotherapy from the context in which the targeted client experiences distress diminishes its effectiveness [60]. Contextual factors must also be taken into consideration when using MABI in varied settings [61]. Our findings show that low-resource settings have structural barriers that must be considered when introducing interventions, congruent with previous findings which underscore the importance of addressing accessibility, affordability and constraints if an intervention is to be sustained in a limited-resource setting [48].

Similarly, while other findings have suggested that altering a well-developed intervention risks jeopardizing the extent to which it remains true to the scientific principles that have guided the development of the intervention [62], other framework adaptations have shown that it is possible to adapt interventions while keeping the core tenets intact [63]. Additionally,

models of intervention adaptation posit that retaining its core tenets and altering the modifiable aspects can improve intervention effectiveness in a new context [30]. Thus, contextualizing the modifiable tenets of the DNA-v will result in quicker uptake and meet the need for a stage-appropriate psychosocial intervention needed to improve mental health and facilitate adherence among AWH in Uganda.

Results from the testing phase indicated higher acceptance of the adapted DNA-v among AWH. The most interesting findings included: participants developing confidence to face fears, self-acceptance, making mindful decisions, and normalizing life experiences. This has significant implications for the usefulness of the intervention in meeting its intended need. This is also in line with previous research highlighting self-acceptance and decision-making to be important aspects that need to be addressed when supporting AWH because they affect retention in care and the treatment cascade [64,65]. In addition, participants believed that the intervention resonated with their specific experiences and aligned with their unique needs which motivated them to stay in the study amidst challenges. This aligns with previous research which emphasizes that the appropriateness of the intervention to the specific needs of the targeted users improves retention in care and promotes continuity beyond the study [66].

Relatedly, our study shows that interventions which promote personal meaning and present-moment awareness offer promise for encouraging value-consistent behaviors such as adherence to medical advice. AWH in the study were optimistic that since the DNA-v helps them refocus attention to what is most important in a given moment, it has the potential to promote health consistent behaviors. Our study coincides with previous findings that underscore the relevance of promoting goals and meaning in life when designing strategies to promote adherence to ART [67]. We further have demonstrated how reinforcing self-beliefs can influence choices adolescents make in their lives. This finding implies that interventions designed to support adolescents should prioritize values clarification and developing a renewed sense of self to empower them to navigate daunting life situations. Given that the mental health of AWH is impacted by both individual and broader structural factors such as poverty, access to facilities, and discrimination in societies [68], a multi-level strategy utilizing a combination of interventions is required. We have presented the DNA-v as psychosocial support which is part of the combined strategies aimed to support AWH. Our approach does not necessarily aim to alter the environment in which an adolescent live (as this requires broader efforts), but rather looks at empowering AWH to develop a flexible self that can thrive regardless of what the environment presents. Even though these are only the first preliminary steps, the initial findings are encouraging and correspond with studies that appraise self-management initiatives targeting both emotional and physical wellness as valid for AWH since living with HIV is an ongoing process with social and medical needs [28].

Finally, using a framework to test the acceptability of the adapted DNA-v gave the study ground to explore various implementation aspects that go beyond participants' subjective feelings towards the intervention. This is an important preparatory step for large-scale implementation. This is consistent with previous research that suggests that studying the acceptability of interventions should be extended beyond subjective elements such as satisfaction with the intervention to also include factors that represent barriers and facilitators to implementation [69].

## Strengths and limitations

The participatory approach to adaptation is a strength of our study [12,58]. This included local mental health experts, providers, AWH, spiritual leaders, and DNA-v intervention developers. The variety and iterative nature of our stakeholder engagement ensured that adaptations were

made *with them* rather than *for them*, a key component in the uptake and usage of innovations [70], using a multi-step process for continual feedback. We utilized varied approaches to generate feedback (workshops and interviews), which are lauded as efficient in mitigating 'group think' during stakeholder engagement [70]. Further, combining two adaptation frameworks (EVM and FMAP) provided structure and content, making the adaptation process rigorous and replicable for future researchers interested in adapting similar interventions in varied settings.

We have built on the previous adaptation of behavioral programs into healthcare systems for young people with HIV. In the *Sauti ya vijana* (voice of the youth) program for AWH in Tanzania, running the intervention at the clinic simplified linkages to care [12]. Although extant literature suggests school-based programs are the best for young people, our study found schools to be stigmatizing environments for ALWHA [48].

Our study findings should be considered in light of several limitations. The intervention adaptation was based on feedback from stakeholders who live and mainly practice in Kampala, which is a different context from other regions of Uganda. Thus, modifications based on feedback from people whose practice is in a given geographical and cultural area may not represent all regions. Further adaptations might be needed when scaling up the intervention to ensure transferability to other contexts. To mitigate the possible power dynamics within groups from limiting certain participants' input, each participant was allotted up to 10 minutes of the group discussion to share their input. Power dynamics may have arisen from factors such as varying levels of education, age, and professional experience. Secondly, as testing of the adapted intervention was clinic-based as advocated by AWH and providers during the adaptation phase, whether the services will be used by those most in need is in question. While clinics might be considered safe places, they tend to attract adolescents who are active in care [71] and who will presumably be more adherent to treatment regimens. Consequently, the AWH who do not attend clinics and potentially need the support most, may not get the opportunity to participate.

However, the adaptation framework we have utilized, and the presentation here of the outcome of this process, can guide future modifications of the interventions in other settings. Lastly, our study did not involve parents or guardians in the adaptation as has been emphasized in the literature on adapting interventions for young people [48]. However, as the consolidated guidelines for Uganda encourage AWH above 12 years of age to seek care without parental consent [6], we considered that it may not be suitable to involve parents. Nevertheless, our study serves as a formative phase on which future adaptations and trials can be based. To promote rigor during the research process, we implemented a standardized interview schedule based on the TFA that was pilot tested and adapted. Additionally, we ensured that trainers did not conduct evaluative interviews and that recruitment was guided by the health care practitioners to minimize bias [72].

## Conclusion

We have presented a stepwise approach to the adaptation of a mindfulness and acceptance intervention intended to support ART adherence among AWH. We have also demonstrated how an intervention developed in a resource-rich setting for the general wellness of young people can be culturally and contextually modified to fit the needs of adolescents in a low-resource context living with a highly stigmatized chronic disease. Preliminary results suggest that the culturally adapted intervention has high potential acceptability and feasibility in the Ugandan context. We plan to assess the effectiveness of the adapted intervention in a future study.

## Supporting information

**S1 Text. Interview schedule.**
(DOCX)

**S2 Text. Consolidated criteria for reporting qualitative research checklist.**
(DOCX)

**S1 Data. Minutes for group 1 stakeholders workshop.**
(DOCX)

**S2 Data. Minutes for group 2 stakeholders workshop.**
(DOCX)

**S3 Data. Minutes for group 2 stakeholders workshop.**
(DOCX)

**S4 Data. Minutes for adolescents' workshop.**
(DOCX)

**S5 Data. Minutes for adolescents' workshop.**
(DOCX)

**S6 Data. Minutes for group 3 stakeholders workshop.**
(DOCX)

**S7 Data. Overall adaptation observations.**
(DOCX)

**S8 Data. Review feedback from adolescents.**
(DOCX)

**S9 Data. Adaptation results summarized.**
(DOCX)

## Acknowledgments

The authors wish to thank Louise Hayes, Joseph Ciarrochi and Ann Bailey for permitting us to adapt the DNA-v, the management and staff of both Kisenyi and Kitebi health centers in Kampala for the support and the timely feedback that enabled the completion of the project and the professors at the Clinical Epidemiology Unit, Makerere University. They also thank the members of the research team: Kelly Gonzaga Kyagaba, Joseph Balikuddembe, Jowella Kukumaho, and the project implementation of the Makerere University Behavioral Social Science research group.

## Author Contributions

**Conceptualization:** Khamisi Musanje, Carol S. Camlin, Moses R. Kamya, Monica Getahun, Rosco Kasujja.

**Formal analysis:** Khamisi Musanje, Hope Kirabo.

**Funding acquisition:** Carol S. Camlin, Moses R. Kamya.

**Investigation:** Hope Kirabo, Ross G. White, Rosco Kasujja.

**Methodology:** Khamisi Musanje, Carol S. Camlin, Deborah Louise Sinclair.

**Project administration:** Khamisi Musanje, Hope Kirabo.

**Supervision:** Moses R. Kamya, Wouter Vanderplasschen, Joan Nangendo, John Kiweewa, Ross G. White, Rosco Kasujja.

**Visualization:** Ross G. White.

**Writing – original draft:** Khamisi Musanje.

**Writing – review & editing:** Carol S. Camlin, Wouter Vanderplasschen, Deborah Louise Sinclair, Monica Getahun, Joan Nangendo, John Kiweewa, Ross G. White, Rosco Kasujja.

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
