## [Decision Letter · Decision Letter 0]

2 Nov 2022

PGPH-D-22-01531

Culturally adapting a mindfulness and acceptance-based intervention to support adherence to antiretroviral therapy among adolescents with HIV in Uganda

Dear Khamisi Musanje,

Thank you for submitting your manuscript to PLOS Global Public Health. After careful consideration, we feel that it has merit but does not fully meet PLOS Global Public Health’s publication criteria as it currently stands. Therefore, we invite you to submit a revised version of the manuscript that addresses the points raised during the review process.

Your manuscript shows promise and is a valuable contribution to the research on adapting interventions in a culturally different setting than the original intervention.

Please ensure you proofread, correct spelling and grammar errors before resubmitting.

There are major revisions required before the manuscript can be considered for publication. Notably, how study 1 and study 2 are presented, re-working the results and discussion section. There are many points each reviewer makes that will need to be addressed before you can resubmit. I believe with these revisions your manuscript will be much stronger.

We look forward to receiving your revised manuscript.

Kind regards,

Bonnie Fournier, PhD, RN

Academic Editor

Journal Requirements:

1. Please send a completed 'Competing Interests' statement, including any COIs declared by your co-authors. If you have no competing interests to declare, please state "The authors have declared that no competing interests exist". Otherwise please declare all competing interests beginning with the statement "I have read the journal's policy and the authors of this manuscript have the following competing interests:"

3. Please provide separate figure files in .tif or .eps format only and remove any figures embedded in your manuscript file. Please also ensure that all files are under our size limit of 10MB.

4. We have noticed that you have a list of Supporting Information legends in your manuscript. However, there are no corresponding files uploaded to the submission. Please upload them as separate files with the item type 'Supporting Information'. 

5. In the online submission form, you indicated that "The data that support the findings of this study are available from the corresponding author, [KM], upon reasonable request". All PLOS journals now require all data underlying the findings described in their manuscript to be freely available to other researchers, either 1. In a public repository, 2. Within the manuscript itself, or 3. Uploaded as supplementary information.

Reviewers' comments:

Reviewer's Responses to Questions

**Comments to the Author**

1. Does this manuscript meet PLOS Global Public Health’s publication criteria? Is the manuscript technically sound, and do the data support the conclusions? The manuscript must describe methodologically and ethically rigorous research with conclusions that are appropriately drawn based on the data presented.

Reviewer #1: Partly

Reviewer #2: Yes

2. Has the statistical analysis been performed appropriately and rigorously?

Reviewer #1: N/A

Reviewer #2: N/A

3. Have the authors made all data underlying the findings in their manuscript fully available (please refer to the Data Availability Statement at the start of the manuscript PDF file)?

Reviewer #1: Yes

Reviewer #2: No

4. Is the manuscript presented in an intelligible fashion and written in standard English?

Reviewer #1: Yes

Reviewer #2: Yes

5. Review Comments to the Author

Reviewer #1: Thank you for the opportunity to review this manuscript reporting on two related studies aimed to culturally adapt a mindfulness and acceptance-based intervention for adolescents with HIV (AWH) living in Kampala, Uganda. It is noted that there are few evidence-based psychosocial interventions that are culturally and contextually appropriate for AWH in Uganda. The authors used a community participatory framework when engaging various stakeholders to adapt the intervention. This is a notable strength as it allows to better adapt the intervention for AWH within a specific setting. The authors describe this well in the discussion “… adaptations were made with them rather than for them…”

Nevertheless, several aspects of the introduction and methods sections could use clarification and/or more detail. In addition, results presented from Study 2 are very brief and not described in sufficient detail. Furthermore, the discussion lacks information and does not address broader structural factors that may impact AWH’s mental health and well-being in Uganda. Please see specific questions and comments below in the spirit of constructive feedback.

Note: there a few minor grammar and spelling errors in the manuscript. For instance, line 110, “it’s” should be spelled “its” as the authors are referring to DNA-v; and “percieved effectiveness” should be spelled “perceived effectiveness” in the analysis section of Study 2.

Comments and suggestions:

ABSTRACT -

Line 15: It is not clear who are these stakeholders – please add details (e.g., local mental health experts, spiritual and adolescents with HIV) and how many stakeholders were engaged (n=30).

Line 21: how many adolescents were recruited for qualitative interviews?

Line 24: suggest to say “the adapted intervention was perceived as acceptable among…”

INTRODUCTION –

Lines 96-98: the authors say that research supports the effectiveness of DNA-v among adolescents with chronic conditions – does this include HIV? If not, the introduction should mention this and the need to adapt the DNA-v intervention to adolescents with HIV (in addition to cultural and contextual adaptations).

Line 100: please outline some specific difficulties psychotherapies encounter in non-Western settings (e.g., language, culture, etc.)

Line 103: the authors note that limited studies have adapted evidence-based psychosocial interventions for AWH – where and how have these studies been conducted and how were the interventions adapted? Please clarify.

Lines 111-112: Where will the larger study take place (e.g., in mental health clinics in and surrounding Kampala)?

METHODS -

Lines 114-115: Add information on how participant consent was obtained for both studies. Study 1 mentions written consent, but study 2 does not provide this information.

Lines 148-149: DNA-v was selected as it may improve young people’s mental health. But where and how was this information demonstrated? Add details from the sourced studies – for instance, in what setting and context DNA-v studied? How was young people’s mental health assessed (e.g., self-reported, validated scales, etc)?

STUDY 1 –

Lines 167-168: why were only six components of the EVM used? What is the reason for not including the other 2 components as per the EVM?

Lines 184-185: provide median age and standard deviation rather than average age. The authors state that stakeholders were majority female (n=18) – it is questionable whether 60% (18/30) should be labelled as a “majority”.

Lines 201-212: The authors should provide more detail on how many workshops were conducted and with whom. It is not clear which stakeholders are considered “local mental health providers and experts” – does this include all stakeholders except AWH? Does it include spiritual leaders? Moreover, how long was the workshop with AWH? Was the workshop also recorded, transcribed and translated?

STUDY 2 (line numbers are no longer available) –

Pages 20-21 (procedures): it is not clear whether all 9 AWH who were interviewed attended all six DNA-v sessions. Moreover, how long were the in-depth interviews? Were they conducted one-on-one with a researcher? Can the authors provide examples of questions from their interview guide (if interviews were semi-structured)?

Page 21 (analysis): The description of the analysis is brief. The authors previously mentioned that questions were guided by a theoretical framework – how did this influence the analysis? In stage 3, please clarify what is meant by a “set of themes” – does this relate to a coding scheme (or sets of codes)? Were no new codes added to the coding scheme during analysis of the six remaining interviews?

Page 23 (results): the final sentence of the results describes that the authors synthesized findings and made a final round of refinements, but no other detail is provided. A table of summarized findings and final refinements should be included.

DISCUSSION:

The discussion is limited to the adaptation of the DNA-v to AWH (study 1). However, there is little mention of the “testing” of the DNA-v with 9 AWH participants and results from in-depth interviews.

In addition, the authors should discuss limitations to where and how the DNA-v intervention is delivered. For instance, as ART adherence is sub-optimal in adolescents, those who seek treatment regularly at a health centre may already have better adherence and HIV knowledge than those who do not visit health centre regularly. Therefore, is it possible that those who were included in the 'testing' phase may have better confidence and self-acceptance than those that do not attend clinics and/or treatment regularly? Also, it was mentioned that some faced challenges to attend DNA-v sessions as they were occurring during school days/hours. In light of increased school dropout among AWH, it would be important to suggest ways on how to better support AWH who are facing these challenges.

Finally, the authors should discuss broader structural factors that may impact AWH's mental health and well-being. For instance, self-acceptance and self-stigmatization may be improved if HIV knowledge, acceptance and HIV stigma was addressed amongst young people not living with HIV. Interventions aimed for all students (not only those with HIV) within the school environment may be favorable and create environments that are supportive for those living with HIV.

Reviewer #2: Thanks for the opportunity to review this important work describing a culturally-adapted intervention for improving mental health among yong people in Uganda. The manuscript makes an important contribution to the literature on cultural adaptation methodology and the promise of interventions for young people in SSA. Generally, the manuscript is very well written, and the approach to cultural adaptation is persuasive. In places, there are errors in spelling and syntax - please complete a careful proofread.

Sub-study 2 is less persuasive than Sub-study 1 as an independent element. However, it contributes to the general sense that the adaptation process worked. The overall argument and presentation could be strengthened by integrating the sub-studies with a usual presentation format e.g., methods, results, discussion. As AWH were also interviewed for sub-study 1 and these discussions had important implications for study location e.g., that schools are not good sites for interventions due to stigma, findings from across adaptation discussions and AWH qualitative interviews perhaps could be coded and integrated/reported as part of AWH qualitative findings. This could provide more substantiated support for the results sub-section reporting on actions taken to improve adaptation. Practically this would involve integrating the methods into one section, and presenting the results in a more comprehensive way with qualitative findings first, then adaptation actions second. In its current form, results from AWH coded interviews could be expanded.

Additional suggested revisions:

p.4, line 54. Prior to the statement "more than half of the new infections"... no information was given on the number of new infections globally, in SSA, and in Uganda. It would also make sense to state the prevalence in SSA, and in Uganda overall, and among young people. From here on, the introduction does an excellent job presenting the necessary context.

p.10, line 188. It would be helpful to know how the participants were selected within stakeholder categories e.g., purposive selection, convenience, targeting gender balance or balance of other characteristics or experiences?

p.100, line 215. What was the step between recording of discussions and synthesis e.g., how were themes collected and collated in a uniform way from recordings and transcriptions e.g., coding framework, like the second sub-study?

p. 16. Would translation have to occur into all 41 Ugandan dialects? Given this work is taking place in Kampala, it might have been beneficial to translate/back-translate the manual into the most commonly spoken language within Kampala (e.g., Luganda and/or Swahili) in order to ensure that the semantics were captured. I think the approach taken is nonetheless acceptable. p.17. Without a translated version, adding local words and slang into the manual seemed like an appropriate way to facilitate semantic understanding.

p.20. Participants. Same question as above on participant selection - please clarify the approach.

p.20. Procedure. How were IDIs conducted? Was a topic guide used? If so, please include. If not, please describe the approach. How long did interviews last (time range)?

p.21. I believe this is the first time reading about "acceptability-TFA" and not sure what it is (a search does not bring up a prior use). If this is the major theoretical framework used for analysis, it should be introduced and described in the introduction or methods.

p.21. Results. It would be helpful to introduce this section with a 1-2 lines of overview of the major themes identified. At the moment, the findings presented seem somewhat thin, and could be further extrapolated. Currently, only positive elements are reported. Did participants offer any constructive feedback either relating to their own experience, or perhaps something related to how they perceive broader acceptance of the intervention? Were there outliers, or contrasting views? Did participants report anything relating to their perception that the intervention could help to encourage adherence to ART?

o.23. Discussion. At this stage, it might be helpful here to give some examples of successfully adapted interventions that were also evaluated and demonstrated evidence for effectiveness.

p.24, top. "These findings..." Is this line referring to the present findings (e.g., author's work) or findings from the study cited in the line before? What is meant by "non-dominant cultures?" Does this mean the majority ethnicity?

"Our findings suggest that low-resource settings have structural barriers..." This is absolutely true, but I did not take that message away within the context of the present work from either sets of results presented. If participants discussed such barriers, consider adding these findings to the results section(s) to facilitate this part of the discussion.

p. 24, bottom. The "most interesting findings stated here" are not evident in results. This may reinforce the feedback that the results section for study 2 should be expanded and perhaps re-organized.

p.25, first para. Correct syntax in the last line. Also, rephrase slightly for clarity: "...acceptability to be a multifaceted construct that should be constructed beyond its constructs..."

p.26. "...our study found schools to be stigmatizing environments..." This is an important finding but one that is not presently well substantiated in the presentation of results from adaptation discussions as they are not currently coded and analyzed. It would be helpful to get a sense of the points of agreement within and across stakeholder groups on adaptation actions, and any outlier views, from these discussions.

p.26, bottom. This is an important point on involvement of parents/guardians. It would be helpful in methods to clarify local requirements for consent/assent. For participants who assented, was consent of parent/guardian required? If not, how did assent process work for minors (add to page 10, line 185)

Ref. 37. Is there a URL? Not sure there is enough information provided to track this ref down.

6. PLOS authors have the option to publish the peer review history of their article (what does this mean?). If published, this will include your full peer review and any attached files.

**Do you want your identity to be public for this peer review?** For information about this choice, including consent withdrawal, please see our Privacy Policy.

Reviewer #1: No

Reviewer #2: No

---

## [Decision Letter · Decision Letter 1]

27 Dec 2022

PGPH-D-22-01531R1

Culturally adapting a mindfulness and acceptance-based intervention to support adherence to antiretroviral therapy among adolescents with HIV in Uganda

Dear Khamisi Musanje,

Thank you for submitting your manuscript to PLOS Global Public Health. After careful consideration, we feel that it has merit but does not fully meet PLOS Global Public Health’s publication criteria as it currently stands. Therefore, we invite you to submit a revised version of the manuscript that addresses the points raised during the review process.

Please find reviewer's comments regarding your latest revisions to your manuscript. Ensure you address their concerns and revise your manuscript accordingly in your re-submission. Specifically the title of your manuscript requires revision (see reviewer's comments).

In general, we would expect qualitative studies to include the following: 1) defined objectives or research questions; 2) description of the sampling strategy, including rationale for the recruitment method, participant inclusion/exclusion criteria and the number of participants recruited; 3) detailed reporting of the data collection procedures; 4) data analysis procedures described in sufficient detail to enable replication; 5) a discussion of potential sources of bias; and 6) a discussion of limitations. Since the COREQ criteria was used, please ensure you follow and outline all details laid out in the COREQ checklist.

Please submit your revised manuscript by January 16th, 2023. If you will need more time than this to complete your revisions, please reply to this message or contact the journal office at globalpubhealth@plos.org. Please include the following items when submitting your revised manuscript:

Once again, we look forward to receiving your revised manuscript.

Kind regards,

Bonnie Fournier, PhD, RN

Academic Editor

Journal Requirements:

2. Please provide separate figure files in .tif or .eps format only and remove any figures embedded in your manuscript file. Please also ensure that all files are under our size limit of 10MB.

3. Please upload a copy of Figure 1 which you refer to in your text on page 10. Or, if the figure is no longer to be included as part of the submission please remove all reference to it within the text.

4. We have noticed that you have cited Table 1 in the manuscript file but there are no corresponding tables in the manuscript. Please amend your manuscript to include this table, noting that tables should not be uploaded as individual files.

Reviewers' comments:

Reviewer's Responses to Questions

**Comments to the Author**

1. If the authors have adequately addressed your comments raised in a previous round of review and you feel that this manuscript is now acceptable for publication, you may indicate that here to bypass the “Comments to the Author” section, enter your conflict of interest statement in the “Confidential to Editor” section, and submit your "Accept" recommendation.

Reviewer #1: All comments have been addressed

Reviewer #2: (No Response)

2. Does this manuscript meet PLOS Global Public Health’s publication criteria? Is the manuscript technically sound, and do the data support the conclusions? The manuscript must describe methodologically and ethically rigorous research with conclusions that are appropriately drawn based on the data presented.

Reviewer #1: Partly

Reviewer #2: Yes

3. Has the statistical analysis been performed appropriately and rigorously?

Reviewer #1: N/A

Reviewer #2: N/A

4. Have the authors made all data underlying the findings in their manuscript fully available (please refer to the Data Availability Statement at the start of the manuscript PDF file)?

Reviewer #1: Yes

Reviewer #2: Yes

5. Is the manuscript presented in an intelligible fashion and written in standard English?

Reviewer #1: Yes

Reviewer #2: Yes

6. Review Comments to the Author

Reviewer #1: Thank you for the opportunity to review this revised version of a manuscript reporting on two related studies aimed to culturally adapt a mindfulness and acceptance-based intervention for adolescents with HIV (AWH) living in Kampala, Uganda. The authors have made several revisions and have addressed previous reviewer feedback.

Nonetheless, minor revisions to the manuscript are warranted:

1. Lines 240-246 (Analysis of Phase 1) seems to lack information – did the authors code and analyze transcripts of both group discussions?

2. Lines 292-206 (Analysis of Phase 2): Although the authors state that “data were analyzed using abductive analysis”, the generation of themes seems constrained to a deductive approach. Evidence of an inductive process is lacking.

3. Lines 408-409: The authors report that this phase of the study has been reported in accordance with the COREQ checklist. The COREQ checklist provides guidance for conducting qualitative research. Please complete the checklist in its entirety and include as an attachment to the manuscript.

4. Lines 412-279 (Results of phase 2): This section remains unclear. The authors have added a table presenting themes and sub-themes; however, the written section is not organized by themes. It is recommended to remove Table 2 and present findings similarly to results from phase 1 (i.e., Theme name: description + verbatim quotes from participants).

As a final comment, the authors have made no mention of rigor to demonstrate credibility of qualitative findings. Therefore, key methodological detail is absent and needed to ascertain rigor (e.g., trustworthiness) of this study.

Reviewer #2: Thank-you once again for the opportunity to review this manuscript.

There remains a disconnect between the title, which refers directly to adherence, and the methods/results/conclusion, which are not at all focused on adherence. The title could be revised to remove reference to adherence while highlighting the original purpose of the intervention (e.g., for adolescent mental health). Alternatively, the title could suggest that there are implications for supporting adherence. If the title is to be kept as is, or if adherence is to be kept in the title by suggesting an implication, the first theme/code identified in Phase 2 results (perceived effectiveness/supports adherence) should be expanded to include additional evidence (from interviews i.e., quotations) that reflect the theme. This key finding should also be highlighted in the discussion/conclusion. The abstract should also be revised accordingly.

Additional suggestions below (Using page/line numbers from tracked version).

line 124 Capitalize Theoretical Framework of Acceptability?

line 189. Suggest one methods section with an additional level of sub-heading for phase 1 and phase 2 (e.g., Methods -> Phase 1 ->Cultural adaptation/Participants etc.)

line 210-3. Indicate in brackets the number in each category.

line 279. For possible discussion/consideration - discuss possible power dynamics within groups in limitations and how this was controlled/mitigated

line 308. Attach a copy of the topic guide.

line 327. Rephrase for grammar/clarity.

line 380. Consider deleting "ghetto" which has specific connotations - is this term typically used in the Ugandan context or is "slum" the best word to describe such deprived urban areas in Uganda?

line 388. Not sure why deleted - I thought this was informative.

line 473 (and other places). Usually italics or quotation marks are used, not both. Please check for best practice with editor.

line 594. Adherence to medication, specifically, or adherence to medical advice in general? Specify.

line 620 The discussion could be improved by including an assessment of quality e.g. validity/transferability of the qualitative findings.

As per journal policy, the dataset i.e., transcripts have been included. Please carefully review and confirm that no identifiers are inadvertently included in the transcripts, especially given the small sample size. If the transcripts create a risk of identifying participants, when participation was confidential, you can consider withholding the transcripts and making them available upon request.

7. PLOS authors have the option to publish the peer review history of their article (what does this mean?). If published, this will include your full peer review and any attached files.

**Do you want your identity to be public for this peer review?** For information about this choice, including consent withdrawal, please see our Privacy Policy.

Reviewer #1: No

Reviewer #2: No

---

## [Editor Report · Decision Letter 2]

24 Jan 2023

Culturally adapting a mindfulness and acceptance-based intervention to support the mental health of adolescents on antiretroviral therapy  in Uganda

PGPH-D-22-01531R2

Dear Assistant Lecturer Musanje,

We are pleased to inform you that your manuscript 'Culturally adapting a mindfulness and acceptance-based intervention to support the mental health of adolescents on antiretroviral therapy  in Uganda' has been provisionally accepted for publication in PLOS Global Public Health.

Best regards,

Bonnie Fournier, PhD, RN

Academic Editor
